# CompRess: Self-Supervised Learning by Compressing Representations

**Soroush Abbasi Koohpayegani**\*    **Ajinkya Tejankar**\*    **Hamed Pirsiavash**
University of Maryland, Baltimore County
`{soroush,at6,hpirsiav}@umbc.edu`

## Abstract

Self-supervised learning aims to learn good representations with unlabeled data. Recent works have shown that larger models benefit more from self-supervised learning than smaller models. As a result, the gap between supervised and self-supervised learning has been greatly reduced for larger models. In this work, instead of designing a new pseudo task for self-supervised learning, we develop a model compression method to compress an already learned, deep self-supervised model (teacher) to a smaller one (student). We train the student model so that it mimics the relative similarity between the datapoints in the teacher's embedding space. For AlexNet, our method outperforms all previous methods including the fully supervised model on ImageNet linear evaluation ($59.0\%$ compared to $56.5\%$) and on nearest neighbor evaluation ($50.7\%$ compared to $41.4\%$). To the best of our knowledge, this is the first time a self-supervised AlexNet has outperformed supervised one on ImageNet classification. Our code is available here: https://github.com/UMBCvision/CompRess

## 1  Introduction

Supervised deep learning needs lots of annotated data, but the annotation process is particularly expensive in some domains like medical images. Moreover, the process is prone to human bias and may also result in ambiguous annotations. Hence, we are interested in self-supervised learning (SSL) where we learn rich representations from unlabeled data. One may use these learned features along with a simple linear classifier to build a recognition system with small annotated data. It is shown that SSL models trained on ImageNet without labels outperform the supervised models when transferred to other tasks [9, 24].

Some recent self-supervised learning algorithms have shown that increasing the capacity of the architecture results in much better representations. For instance, for SimCLR method [9], the gap between supervised and self-supervised is much smaller for ResNet-50x4 compared to ResNet-50 (also shown in Figure 1). Given this observation, we are interested in learning better representations for small models by compressing a deep self-supervised model.

In edge computing applications, we prefer to run the model (e.g., an image classifier) on the device (e.g., IoT) rather than sending the images to the cloud. During inference, this reduces the privacy concerns, latency, power usage, and cost. Hence, there is need for rich, small models. Compressing SSL models goes beyond that and reduces the privacy concerns at the training time as well. For instance, one can download a rich self-supervised MobileNet model that can generalize well to other tasks and finetune it on his/her own data without sending any data to the cloud for training.

Since we assume our teacher has not seen any labels, its output is an embedding rather than a probability distribution over some categories. Hence, standard model distillation methods [26] cannot

---

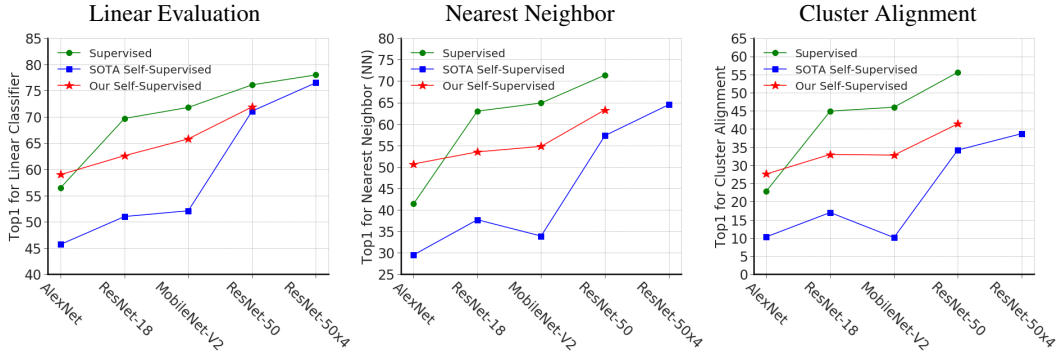

Figure 1: **ImageNet Evaluation:** We compare "Ours-1q" self-supervised model with supervised and SOTA self-supervised models on ImageNet using linear classification (left), nearest neighbor (middle) and cluster alignment (right) evaluations. Our AlexNet model outperforms the supervised counterpart on all evaluations. This model is compressed from ResNet-50x4 trained with SimCLR method using unlabeled ImageNet. All models have seen ImageNet images only. All SOTA SSL models are MoCo except ResNet50x4 that is SimCLR. The teacher for our AlexNet and ResNet50 is SimCLR ResNet50x4 and for ResNet18 and MobileNet-V2 is MoCo ResNet50.

be used directly. One can employ a nearest neighbor classifier in the teacher space by calculating distances between an input image (query) and all datapoints (anchor points) and then converting them to probability distribution. Our idea is to transfer this probability distribution from the teacher to the student so that for any query point, the student matches the teacher in the ranking of anchor points.

Traditionally, most SSL methods are evaluated by learning a linear classifier on the features to perform a downstream task (e.g., ImageNet classification). However, this evaluation process is expensive and has many hyperparameters (e.g., learning rate schedule) that need to be tuned as one set of parameters may not be optimal for all SSL methods. We believe a simple nearest neighbor classifier, used in some recent works [57, 67, 60], is a better alternative as it has no parameters, is much faster to evaluate, and still measures the quality of features. Hence, we use this evaluation extensively in our experiments. Moreover, inspired by [30], we use another related evaluation by measuring the alignment between k-means clusters and image categories.

Our extensive experiments show that our compressed SSL models outperform state-of-the-art compression methods as well as state-of-the-art SSL counterparts using the same architecture on most downstream tasks. Our AlexNet model, compressed from ResNet-50x4 trained with SimCLR method, outperforms standard supervised AlexNet model on linear evaluation (by 2 point), in nearest neighbor (by 9 points), and in cluster alignment evaluation (by 4 points). This is interesting as all parameters of the supervised model are already trained on the downstream task itself but the SSL model and its teacher have seen only ImageNet without labels. To the best of our knowledge, this is the first time an SSL model performs better than the supervised one on the ImageNet task itself instead of transfer learning settings.

## 2 Related work

**Self-supervised learning:** In self-supervised learning for images, we learn rich features by solving a pretext task that needs unlabeled data only. The pseudo task may be colorization [64], inpainting [42], solving Jigsaw puzzles [36], counting visual primitives [37], and clustering images [7].

**Contrastive learning:** Our method is related to contrastive learning [23, 39, 27, 67, 4, 49, 25] where the model learns to contrast between the positive and lots of negative pairs. The positive pair is from the same image and model but different augmentations in [24, 9, 51] and from the same image and augmentation but different models (teacher and student) in [50]. Our method uses soft probability distribution instead of positive/negative classification [67], and does not couple the two embeddings (teacher and student) directly [50] in "Ours-2q" variant. Contrastive learning is improved with a more robust memory bank in [24] and with temperature and better image augmentations in [9]. Our ideas are related to exemplar CNNs [15, 34], but used for compression.

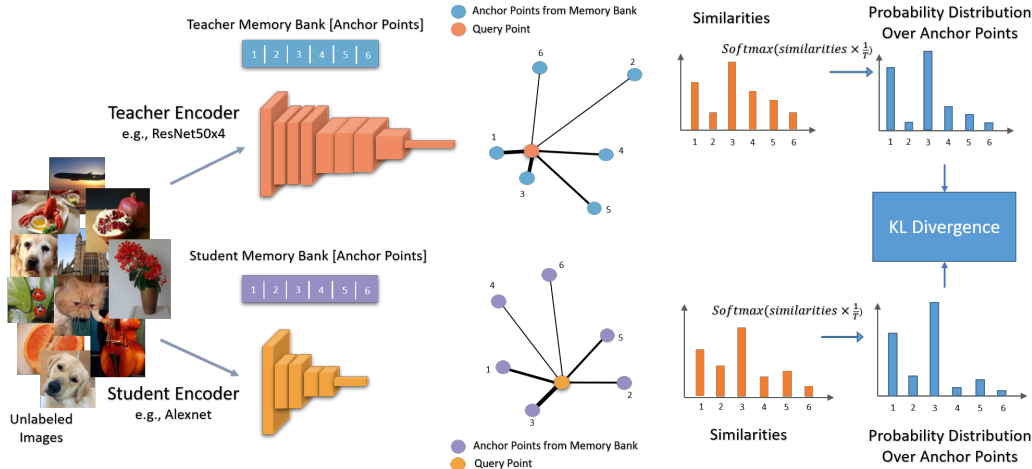

Figure 2: **Our compression method:** The goal is to transfer the knowledge from the self-supervised teacher to the student. For each image, we compare it with a random set of data points called anchors and obtain a set of similarities. These similarities are then converted into a probability distribution over the anchors. This distribution represents each image in terms of its nearest neighbors. Since we want to transfer this knowledge to the student, we get the same distribution from the student as well. Finally, we train the student to minimize the KL divergence between the two distributions. Intuitively, we want each data point to have the same neighbors in both teacher and student embeddings. This illustrates Ours-2q method. For Ours-1q, we simply remove the student memory bank and use the teacher's anchor points for the student as well.

**Model compression:** The task of training a simpler student to mimic the output of a complex teacher is called model compression in [6] and knowledge distillation in [26]. In [26], the softened class probabilities from the teacher are transferred to the student by reducing KL divergence. The knowledge in the hidden activations of intermediate layers of the teacher is transferred by regressing linear projections [45], aggregation of feature maps [62], and gram matrices [59]. Also, knowledge at the final layer can be transferred in different ways [3, 26, 31, 41, 43, 40, 53, 2, 50, 61, 56, 5, 18, 48]. In [2, 50] distillation is formulated as maximization of information between teacher and student.

**Similarity-based distillation:** Pairwise similarity based knowledge distillation has been used along with supervised teachers. [43, 53, 40] use supervised loss in distillation. [41] is probably the closest to our setting which does not use labels in the distillation step. We are different as we use memory bank and SoftMax along with temperature, and also apply that to compressing self-supervised models in large scale. We compare with a reproduced variant of [41] in the experiments (Section 4.3).

**Model compression for self-supervision:** Standard model compression techniques either directly use the output of supervised training [26] or have a supervised loss term [50, 40] in addition to the compression loss term. Thus, they cannot be directly applied to compress self-supervised models. In [38], the knowledge from the teacher is transferred to the student by first clustering the embeddings from teacher and then training the student to predict the cluster assignments. In [58], the method of [38] is applied to regularize self-supervised models.

## 3  Method

Our goal is to train a deep model (e.g. ResNet-50) using an off-the-shelf self-supervised learning algorithm and then, compress it to a less deep model (e.g., AlexNet) while preserving the discriminative power of the features. Figure 2 shows our method.

Assuming a frozen teacher embedding $t(x) \in R^N$ with parameters $\theta_t$ that maps an image $x$ into an $N$-D feature space, we want to learn the student embedding $s(x) \in R^M$ with parameters $\theta_s$ that mimics the same behavior as $t(x)$ if used for a downstream supervised task e.g., image classification. Note that the teacher and student may use architectures from different families, so we do not necessarily want to couple them together directly. Hence, we transfer the similarity between data points from the teacher to the student rather than their final prediction.

For simplicity, we use $t_i = t(x_i)$ for the embedding of the model $t(x)$ on the input image $x_i$ normalized by $\ell_2$ norm. We assume a random set of the training data $\{x_j\}_{j=1}^n$ are the *anchor* points and embed them using both teacher and student models to get $\{t_j^a\}_{j=1}^n$ and $\{s_j^a\}_{j=1}^n$. Given a *query* image $q_i$ and its embeddings $t_i^q$ for teacher and $s_i^q$ for student, we calculate the pairwise similarity between $t_i^q$ and all anchor embeddings $\{t_j^a\}_{j=1}^n$, and then optimize the student model so that in the student's embedding space, the query $s_i^q$ has the same relationship with the anchor points $\{s_j^a\}_{j=1}^n$.

To measure the relationship between the query and anchor points, we calculate their cosine similarity. We convert the similarities to the form of a probability distribution over the anchor points using SoftMax operator. For the teacher, the probability of the $i$-th query for the $j$-th anchor point is:

$$p_j^i(t) = \frac{\exp(t_i^{qT} t_j^a / \tau)}{\sum_{k=1}^n \exp(t_i^{qT} t_k^a / \tau)}$$

where $\tau$ is the temperature hyperparamater. Then, we define the loss for a particular query point as the $KL$ divergence between the probabilities over all anchor points under the teacher and student models, and we sum this loss over all query points:

$$L(t, s) = \sum_i KL(p^i(t)||p^i(s))$$

where $p^i(s)$ is the probability distribution of query $i$ over all anchor points on the student network. Finally, since the teacher is frozen, we optimize the student by solving:

$$\arg\min_{\theta_s} L(t, s) = \arg\min_{\theta_s} \sum_{i,j} -p_j^i(t).log(p_j^i(s))$$

**Memory bank:** One may use the same minibatch for both query and anchor points by excluding the query from each set of anchor points. However, we need a large set of anchor points (ideally the whole training set) so that they have large variation to cover the neighborhood of any query image. Our experiments verify that using the minibatch of size 256 for anchor points is not enough for learning rich representations. This is reasonable as ImageNet has 1000 categories so the query may not be close to any anchor point in a minibatch of size 256. However, it is computationally expensive to process many images in a single iteration due to limited computation and memory. Similar to [57], we maintain a memory bank of anchor points from several most recent iterations. We use momentum contrast [24] framework for implementing memory bank for the student. However, unlike [24], we find that our method is not affected by the momentum parameter which requires further investigation. Since the teacher is frozen, we implement its memory bank as a simple FIFO queue.

**Temperature parameter:** Since the anchor points have large variation covering the whole dataset, many of them may be very far from the query image. We use a small temperature value (less than one) since we want to focus mainly on transferring the relationships from the close neighborhoods of the query rather than faraway points. Note that this results in sharper probabilities compared to $\tau = 1$. We show that $\tau = 1$ degrades the results dramatically. The temperature value acts similarly to the kernel width in kernel density estimation methods.

**Student using teacher's memory bank:** So far, we assumed that the teacher and student embeddings are decoupled, so used a separate memory bank (queue) for each. We call this method **Ours-2q**. However, we may use the teacher's anchor points in calculating the similarity for the student model. This way, the model may learn faster and be more stable in the initial stages of learning, since the teacher anchor points are already mature. We call this variation **Ours-1q** in our experiments. Note that in "Ours-1q" method, we do not use momentum since the teacher is constant.

**Caching the teacher embeddings:** Since we are interested in using very deep models (e.g., ResNet-50x4) as the teacher, calculating the embeddings for the teacher is expensive in terms of both computation and memory. Also, we are not optimizing the teacher model. Hence, for such large models, we can cache the results of the teacher on all images of the dataset and keep them in the memory. This caching has a drawback that we cannot augment the images for the teacher, meaning that the teacher sees exact same images in all epochs. However, since the student still sees augmented images, it is less prone to overfitting. On the other hand, this caching may actually help the student by encouraging the relationship between the query and anchor points to be close even under different augmentations, hence, improving the representation in a way similar to regular contrastive learning

[57, 24]. In our experiments, we realize that caching degrades the results by only a small margin while is much faster and efficient in learning. We use caching when we compress from ResNet-50x4 to AlexNet.

# 4 Experiments and results

We use different combinations of architectures as student-teacher pairs (listed in Table 1). We use three teachers : (a) ResNet-50 model which is trained using MoCo-v2 method for 800 epochs [10], (b) ResNet-50 trained with SwAV [8] for 800 epochs, and (c) ResNet-50x4 model which is trained using SimCLR method for 1000 epochs [9]. We use the officially published weights of these models [44, 47, 54]. For supervised models, we use the official PyTorch weights [52]. We use ImageNet (ILSVRC2012) [46] without labels for all self-supervised and compression methods, and use various datasets (ImageNet, PASCAL-VOC [16], Places [66], CUB200 [55], and Cars196 [32]) for evaluation.

**Implementation details:** Here, we report the implementation details for Ours-2q and Ours-1q compression methods. The implementation details for all baselines and transfer experiments are included in the appendix. We use PyTorch along with SGD (weight decay=$1e-4$, learning rate=0.01, momentum=0.9, epochs=130, and batch size=256). We multiply learning rate by 0.2 at epochs 90 and 120. We use standard ImageNet data augmentation found in PyTorch. Compressing from ResNet-50x4 to ResNet-50 takes ~100 hours on four Titan-RTX GPUs while compressing from ResNet-50 to ResNet-18 takes ~90 hours on two 2080-TI GPUs. We adapt the unofficial implementation of MoCo [24] in [11] to implement memory bank for our method. We use memory bank size of $128,000$ and set moving average weight for key encoder to 0.999. We use the temperature of 0.04 for all experiments involving SimCLR ResNet-50x4 and MoCo ResNet-50 teachers. We pick these values based on the ablation study done for the temperature parameter in Section 4.5. For SwAV ResNet-50 teacher, we use a temperature of 0.007 since we find that it works better than 0.04.

## 4.1 Evaluation Metrics

**Linear classifier (Linear):** We treat the student as a frozen feature extractor and train a linear classifier on the labeled training set of ImageNet and evaluate it on the validation set with Top-1 accuracy. To reduce the computational overhead of tuning the hyperparameters per experiment, we standardize the Linear evaluation as following. We first normalize the features by $\ell_2$ norm, then shift and scale each dimension to have zero mean and unit variance. For all linear layer experiments, we use SGD with lr=0.01, epochs=40, batch size=256, weight decay=$1e-4$, and momentum=0.9. At epochs 15 and 30, the lr is multiplied by 0.1.

**Nearest Neighbor (NN):** We also evaluate the student representations using nearest neighbor classifier with cosine similarity. We use FAISS GPU library [1] to implement it. This method does not need any parameter tuning and is very fast (~25 minutes for ResNet-50 on a single 2080-TI GPU)

**Cluster Alignment (CA):** The goal is to measure the alignment between clusters of our SSL representations with visual categories, e.g., ImageNet categories. We use k-means (with $k$=1000) to cluster our self-supervised features trained on unlabeled ImageNet, map each cluster to an ImageNet category, and then evaluate on ImageNet validation set. In order to map clusters to categories, we first calculate the alignment between all (cluster- category) pairs by calculating the number of common images divided by the size of cluster. Then, we find the best mapping between clusters and categories using Hungarian algorithm [33] that maximizes total alignment. This labels the clusters. Then, we report the classification accuracy on the validation set. This setting is similar to the object discovery setting in [30]. In Figure 3 (c), we show some random images from random clusters where images inside each cluster are semantically related.

## 4.2 Baselines:

**Contrastive Representation Distillation (CRD):** CRD [50] is an information maximization based SOTA method for distillation that includes a supervised loss term. It directly compares the embeddings of teacher and student as in a contrastive setting. We remove the supervised loss in our experiments.

**Cluster Classification (CC):** Cluster Classification [38] is an unsupervised knowledge distillation method that improves self-supervised learning by quantizing the teacher representations. This is similar to the recent work of ClusterFit [58].

Table 1: **Comparison of distillation methods on full ImageNet:** Our method is better than all compression methods for various teacher-student combinations and evaluation benchmarks. In addition, as reported in Table 5 and Figure 1, when we compress ResNet-50x4 to AlexNet, we get **59.0%** for Linear, **50.7%** for Nearest Neighbor (NN), and **27.6%** for Cluster Alignment (CA) which outperforms the supervised model. On NN, our ResNet-50 is only 1 point worse than its ResNet-50x4 teacher. Note that models below the teacher row use the student architecture. Since a forward pass through the teacher is expensive for ResNet50x4, we do not compare with CRD, Reg, and Reg-BN.

| Teacher<br>Student | MoCo ResNet-50<br>AlexNet | | | MoCo ResNet-50<br>ResNet-18 | | | MoCo ResNet-50<br>MobileNet-V2 | | | ResNet-50x4<br>ResNet-50 | | |
|---|---|---|---|---|---|---|---|---|---|---|---|---|
| | Linear | NN | CA | Linear | NN | CA | Linear | NN | CA | Linear | NN | CA |
| Teacher | 70.8 | 57.3 | 34.2 | 70.8 | 57.3 | 34.2 | 70.8 | 57.3 | 34.2 | 75.6 | 64.5 | 38.7 |
| Supervised | 56.5 | 41.4 | 22.9 | 69.8 | 63.0 | 44.9 | 71.9 | 64.9 | 46.0 | 76.2 | 71.4 | 55.6 |
| CC [38] | 46.4 | 31.6 | 13.7 | 61.1 | 51.1 | 25.2 | 59.2 | 50.2 | 24.7 | 68.9 | 55.6 | 26.4 |
| CRD [50] | 54.4 | 36.9 | 14.1 | 58.4 | 43.7 | 17.4 | 54.1 | 36.0 | 12.0 | - | - | - |
| Reg | 49.9 | 35.6 | 9.5 | 52.2 | 41.7 | 25.6 | 48.0 | 38.6 | 25.4 | - | - | - |
| Reg-BN | 56.1 | 42.8 | 22.3 | 58.2 | 47.3 | 27.2 | 62.3 | 48.7 | 27.0 | - | - | - |
| Ours-2q | 56.4 | **48.4** | **33.3** | 61.7 | 53.4 | **34.7** | 63.0 | 54.4 | **35.5** | 71.0 | 63.0 | 41.1 |
| Ours-1q | **57.5** | 48.0 | 27.0 | **62.6** | 53.5 | 33.0 | **65.8** | 54.8 | 32.8 | **71.9** | 63.3 | **41.4** |

Table 2: **Comparison of distillation methods on full ImageNet for SwAV ResNet-50 (teacher) to ResNet-18 (student):**. Note that SwAV (concurrent work) [8] is different from MoCo and SimCLR in that it performs contrastive learning through online clustering.

| Method | Linear | NN | CA |
|---|---|---|---|
| Teacher | 75.6 | 60.7 | 27.6 |
| Supervised | 69.8 | 63.0 | 44.9 |
| CRD | 58.2 | 44.7 | 16.9 |
| CC | 60.8 | 51.0 | 22.8 |
| Reg-BN | 60.6 | 47.6 | 20.8 |
| Ours-2q | 62.4 | 53.7 | **26.7** |
| Ours-1q | **65.6** | **56.0** | 26.3 |

Table 3: **NN evaluation for ImageNet with fewer labels:** We report NN evaluation on validation data using small training data (both ImageNet) for ResNet-18 compressed from MoCo ResNet-50. For 1-shot, we report the standard deviation over 10 runs.

| Model | 1-shot | 1% | 10% |
|---|---|---|---|
| Supervised (entire labeled ImageNet) | 29.8 ($\pm$0.3) | 48.5 | 56.8 |
| CC [38] | 16.3 ($\pm$0.3) | 31.6 | 41.9 |
| CRD [50] | 11.4 ($\pm$0.3) | 23.3 | 33.6 |
| Reg-BN | 21.5 ($\pm$0.1) | 33.4 | 40.1 |
| Ours-2q | **29.0** ($\pm$0.3) | **41.2** | **47.6** |
| Ours-1q | 26.5 ($\pm$0.3) | 39.6 | 47.2 |

**Regression (Reg):** We implement a modified version of [45] that regresses only the embedding layer features [61]. Similar to [45, 61], we add a linear projection head on top of the student to match the embedding dimension of the teacher. As noted in CRD [50], transferring knowledge from all intermediate layers does not perform well since the teacher and student may have different architecture styles. Hence, we use the regression loss only for the embedding layer of the networks.

**Regression with BatchNorm (Reg-BN):** We realized that Reg does not perform well for model compression. We suspect the reason is the mismatch between the embedding spaces of the teacher and student networks. Hence, we added a non-parametric Batch Normalization layer for the last layer of both student and teacher networks to match their statistics. The BN layer uses statistics from the current minibatch only (element-wise whitening). Interestingly, this simple modified baseline is better than other sophisticated baselines for model compression.

## 4.3 Experiments Comparing Compression Methods

**Evaluation on full ImageNet:** We train the teacher on unlabeled ImageNet, compress it to the student, and evaluate the student using ImageNet validation set. As shown in Table 1, our method outperforms other distillation methods on all evaluation benchmarks. For a fair comparison, on ResNet-18, we trained MoCo for 1,000 epochs and got $54.5\%$ in Linear and $41.1\%$ in NN which does not still match our model. Also, a variation of our method (MoCo R50 to R18) without *SoftMax*,

Table 4: **Transfer to CUB200 and Cars196:** We train the features on unlabeled ImageNet, freeze the features, and return top $k$ nearest neighbors based on cosine similarity. We evaluate the recall at different $k$ values (1, 2, 4, and 8) on the validation set.

| Method AlexNet | Teacher | CUB200 | | | | Cars196 | | | |
|---|---|---|---|---|---|---|---|---|---|
| | | R@1 | R@2 | R@4 | R@8 | R@1 | R@2 | R@4 | R@8 |
| Sup. on ImageNet | - | 33.5 | 45.5 | 59.2 | 71.9 | 26.6 | 36.3 | 45.9 | 57.8 |
| CRD [50] | ResNet-50 | 16.6 | 25.9 | 36.3 | 48.7 | 20.9 | 28.2 | 37.7 | 48.9 |
| Reg-BN | ResNet-50 | 16.8 | 25.5 | 36.2 | 48.0 | 20.9 | 29.0 | 38.5 | 49.7 |
| CC | ResNet-50 | **23.2** | 32.5 | **45.1** | **58.2** | **23.7** | 31.4 | 41.1 | 52.4 |
| Ours-2q | ResNet-50 | 23.1 | **33.0** | **45.1** | 58.0 | 23.6 | **32.8** | **42.9** | **54.9** |
| Ours-1q | ResNet-50 | 22.7 | 31.9 | 43.2 | 55.8 | 22.5 | 30.6 | 40.4 | 52.3 |
| CC | ResNet-50x4 | 23.6 | 33.6 | 44.9 | 58.4 | 25.4 | 33.2 | 43.2 | 54.3 |
| Ours-2q | ResNet-50x4 | **26.5** | **37.0** | **49.4** | **62.4** | **28.4** | **38.5** | **48.7** | **60.4** |
| Ours-1q | ResNet-50x4 | 21.9 | 32.4 | 43.2 | 55.9 | 25.0 | 34.2 | 45.1 | 57.3 |

temperature, and memory bank (similar to [41]) results in $53.6\%$ in Linear and $42.3\%$ in NN. To evaluate the effect of the teacher's SSL method, in Table 2, we use SwAV ResNet-50 as the teacher and compress it to ResNet-18. We still get better accuracy compared to other distillation methods.

**Evaluation on smaller ImageNet:** We evaluate our representations by a NN classifier using only $1\%$, $10\%$, and only 1 sample per category of ImageNet. The results are shown in Table 3. For 1-shot, "Ours-2q" model achieves an accuracy close to the supervised model which has seen all labels of ImageNet in learning the features.

**Transfer to CUB200 and Cars196:** We transfer AlexNet student models to the task of image retrieval on CUB200 [55] and Cars196 [32] datasets. We evaluate on these tasks without any fine-tuning. The results are shown in Table 4. Surprisingly, for the combination of Cars196 dataset and ResNet-50x4 teacher, our model even outperforms the ImageNet supervised model. Since in "Ours-2q", the student embedding is less restricted and does not follow the teacher closely, the student may generalize better compared to "Ours-1q" method. Hence, we see better results for "Ours-2q" on almost all transfer experiments. This effect is similar to [38, 58].

## 4.4 Experiments Comparing Self-Supervised Methods

**Evaluation on ImageNet:** We compare our features with SOTA self-supervised learning methods on Table 5 and Figure 1. Our method outperforms all baselines on all small capacity architectures (AlexNet, MobileNet-V2, and ResNet-18). On AlexNet, it outperforms even the supervised model. Table 6 shows the results of linear classifier using only $1\%$ and $10\%$ of ImageNet for ResNet-50.

**Transferring to Places:** We evaluate our intermediate representations learned from unlabeled ImageNet on Places scene recognition task. We train linear layers on top of intermediate representations similar to [21]. Details are in the appendix. The results are shown in Table 5. We find that our best layer performance is better than that of a model trained with ImageNet labels.

**Transferring to PASCAL-VOC:** We evaluate AlexNet compressed from ResNet-50x4 on PASCAL-VOC classification and detection tasks in Table 7. For classification task, we only train a linear classifier on top of frozen backbone which is in contrast to the baselines that finetune all layers. For object detection, we use the Fast-RCNN [20] as used in [38, 19] to finetune all layers.

## 4.5 Ablation Study

To speed up the ablation study, we use 25% of ImageNet (randomly sampled ~320k images) and cached features of MoCo ResNet-50 as a teacher to train ResNet-18 student. For temperature ablation study, the memory bank size is 128k and for memory bank ablation study, the temperature is 0.04. All ablations were performed with "Ours-2q" method.

**Temperature:** The results of varying temperature between 0.02 and 1.0 are shown in Figure 3(a). We find that the optimal temperature is 0.04, and the student gets worse as the temperature gets

Table 5: **Linear evaluation on ImageNet and Places:** Comparison with SOTA self-supervised methods. We pick the best layer to report the results that is written in parenthesis: 'f7' refers to 'fc7' layer and 'c4' refers to 'conv4' layer. R50x4 refers to the teacher that is trained with SimCLR and R50 to the teacher trained with MoCo. On ResNet-50, our model, that is compressed from SimCLR R50x4, is better than SimCLR itself, but worse than SwAV, BYOL, and InfoMin which are concurrent works. ∗ refers to 10-crop evaluation. † denotes concurrent methods.

| Method | Ref | ImageNet top-1 | Places top-1 |
|---|---|---|---|
| AlexNet | | | |
| Sup. on ImageNet | - | 56.5 (f7) | 39.4 (c4) |
| Inpainting [42] | [60] | 21.0 (c3) | 23.4 (c3) |
| BiGAN [14] | [38] | 29.9 (c4) | 31.8 (c3) |
| Colorization [64] | [19] | 31.5 (c4) | 30.3 (c4) |
| Context [13] | [19] | 31.7 (c4) | 32.7 (c4) |
| Jigsaw [36] | [19] | 34.0 (c3) | 35.0 (c3) |
| Counting [37] | [38] | 34.3 (c3) | 36.3 (c3) |
| SplitBrain [65] | [38] | 35.4 (c3) | 34.1 (c4) |
| InstDisc [57] | [57] | 35.6 (c5) | 34.5 (c4) |
| CC+Vgg+Jigsaw [38] | [38] | 37.3 (c3) | 37.5 (c3) |
| RotNet [19] | [19] | 38.7 (c3) | 35.1 (c3) |
| Artifact [29] | [17] | 38.9 (c4) | 37.3 (c4) |
| AND [28] | [60] | 39.7 (c4) | - |
| DeepCluster [7] | [7] | 39.8 (c4) | 37.5 (c4) |
| LA* [67] | [67] | 42.4 (c5) | 40.3 (c4) |
| CMC [49] | [60] | 42.6 (c5) | - |
| AET [63] | [60] | 44.0 (c3) | 37.1 (c3) |
| RFDecouple [17] | [17] | 44.3 (c5) | 38.6 (c5) |
| SeLa+Rot+aug [60] | [60] | 44.7 (c5) | 37.9 (c4) |
| MoCo | - | 45.7 (f7) | 36.6 (c4) |
| Ours-2q (from R50x4) | - | 57.6 (f7) | **40.4** (c5) |
| Ours-1q (from R50x4) | - | **59.0** (f7) | 40.3 (c5) |

| Method | Ref | ImageNet top-1 |
|---|---|---|
| ResNet-18 | | |
| Sup. on ImageNet | - | 69.8 (L5) |
| InstDisc[57] | [57] | 44.5 (L5) |
| LA* [67] | [67] | 52.8 (L5) |
| MoCo | - | 54.5 (L5) |
| Ours-2q (from R50) | - | 61.7 (L5) |
| Ours-1q (from R50) | - | **62.6** (L5) |
| ResNet-50 | | |
| Sup. on ImageNet | - | 76.2 (L5) |
| InstDisc [57] | [57] | 54.0 (L5) |
| CF-Jigsaw [58] | [58] | 55.2 (L4) |
| CF-RotNet [58] | [58] | 56.1 (L4) |
| LA * [67] | [67] | 60.2 (L5) |
| SeLa [60] | [60] | 61.5 (L5) |
| PIRL [35] | [35] | 63.6 (L5) |
| SimCLR [9] | [9] | 69.3 (L5) |
| MoCo [10] | [10] | 71.1 (L5) |
| InfoMin† [51] | [51] | 73.0 (L5) |
| BYOL† [22] | [22] | 74.3 (L5) |
| SwAV† [8] | [8] | **75.3** (L5) |
| Ours-2q (from R50x4) | - | 71.0 (L5) |
| Ours-1q (from R50x4) | - | 71.9 (L5) |

Table 6: **Evaluation of ResNet-50 features on smaller set of ImageNet:** ResNet-50x4 is used as the teacher. Unlike other methods that fine-tune the whole network, we only train the last layer. Interestingly, despite fine-tuning fewer parameters, our method achieves better results on the 1% dataset. This demonstrates that our method can produce more data-efficient models. ∗ denotes concurrent methods.

| Method | Top-1 1% | Top-1 10% | Top-5 1% | Top-5 10% |
|---|---|---|---|---|
| Supervised | 25.4 | 56.4 | 48.4 | 80.4 |
| InstDisc [57] | - | - | 39.2 | 77.4 |
| PIRL [35] | - | - | 57.2 | 83.8 |
| SimCLR [9] | 48.3 | 65.6 | 75.5 | 87.8 |
| BYOL* [22] | 53.2 | 68.8 | 78.4 | 89.0 |
| SwAV* [8] | 53.9 | **70.2** | 78.5 | **89.9** |
| *Only the linear layer is trained.* | | | | |
| Ours-2q | 57.8 | 66.3 | 80.4 | 87.0 |
| Ours-1q | **59.7** | 67.0 | **82.3** | 87.5 |

Table 7: **Transferring to PASCAL-VOC classification and detection tasks:** All models use AlexNet and ours is compressed from ResNet-50x4. Our model is on par with ImageNet supervised model. For classification, we denote the fine-tuned layers in the parenthesis. For detection, all layers are fine-tuned. ∗ denotes bigger AlexNet [60].

| Method | Cls. mAP | Det. mAP |
|---|---|---|
| Supervised on ImageNet | 79.9 (all) | 59.1 |
| Random Rescaled [60] | 56.6 (all) | 45.6 |
| Context* [13] | 65.3 (all) | 51.1 |
| Jigsaw [36] | 67.6 (all) | 53.2 |
| Counting [37] | 67.7 (all) | 51.4 |
| CC+vgg-Jigsaw++ [38] | 72.5 (all) | 56.5 |
| Rotation [19] | 73.0 (all) | 54.4 |
| DeepCluster* [7] | 73.7 (all) | 55.4 |
| RFDecouple* [17] | 74.7 (all) | 58.0 |
| SeLa+Rot* [60] | 77.2 (all) | 59.2 |
| MoCo [24] | 71.3 (fc8) | 55.8 |
| Ours-2q | **79.7** (fc8) | 58.1 |
| Ours-1q | 76.2 (fc8) | **59.3** |

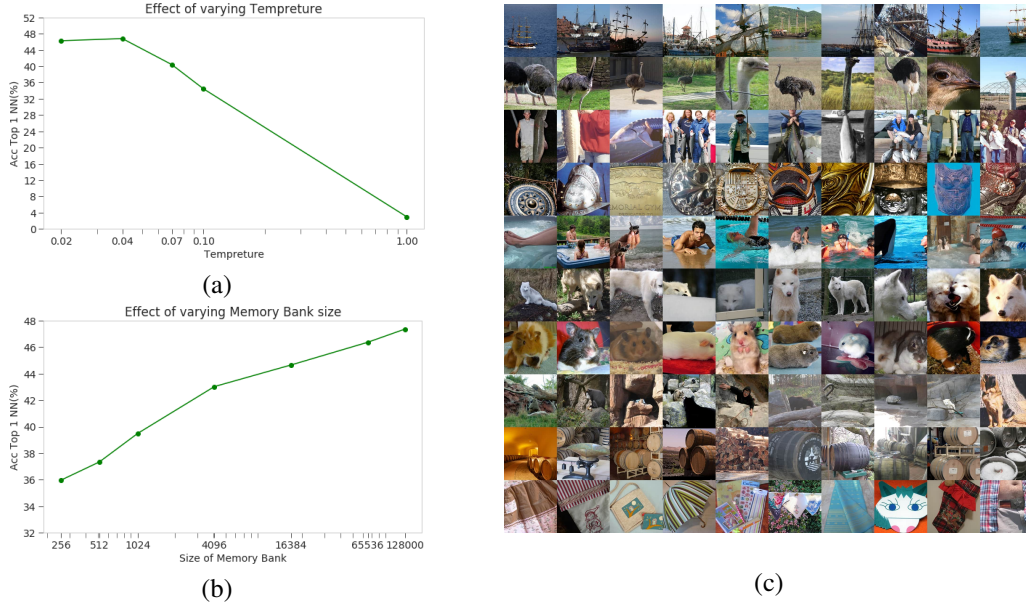

(a)

(b)

(c)

Figure 3: **Ablation and qualitative results:** We show the effect of varying the temperature in **(a)** and memory bank size in **(b)** using ResNet-18 distilled from cached features of MoCo ResNet-50. In **(c)**, we show randomly selected images from randomly selected clusters for our best AlexNet model. Each row is a cluster. This is done *without cherry-picking* or manual inspection. Note that most rows are aligned with semantic categories. We have more of these examples in the Appendix.

closer to 1.0. We believe this happens since a small temperature focuses on close neighborhoods by sharpening the probability distribution. A similar behavior is also reported in [9]. As opposed to the other similarity based distillation methods [41, 43, 40, 53], by using small temperature, we focus on the close neighborhood of a data point which results in an improved student.

**Size of memory bank:** Intuitively, larger number of anchor points should capture more details about the geometry of the teacher's embedding thus resulting in a student that approximates the teacher more closely. We validate this in Figure 3(b) where a larger memory bank results in a more accurate student. When coupled with a small temperature, the large memory bank can help find anchor points that are closer to a query point, thus accurately depicting its close neighborhood.

**Effect of momentum parameter:** We evaluate various momentum parameters [24] in range (0.999, 0.7, 0.5, 0) and got NN accuracy of (47.35%, 47.45%, 47.40%, 47.34%) respectively. It is interesting that unlike [24], we do not see any reduction in accuracy by removing the momentum. The cause deserves further investigation. Note that momentum is only applicable in case of "ours-2q" method.

**Effect of caching the teacher features:** We study the effect of caching the feature of the whole training data in compressing ResNet-50 to ResNet-18 using all ImageNet training data. We realize that caching reduces the accuracy by only a small margin 53.4% to 53.0% on NN and 61.7% to 61.2% on linear evaluation while reducing the running time by a factor of almost 3. Hence, for all experiments using ResNet-50x4, we cache the teacher as we cannot afford not doing so.

## 5   Conclusion

We introduce a simple compression method to train SSL models using deeper SSL teacher models. Our model outperforms the supervised counterpart in the same task of ImageNet classification. This is interesting as the supervised model has access to strictly more information (labels). Obviously, we do not conclude that our SSL method works better than supervised models "in general". We simply compare with the supervised AlexNet that is trained with cross-entropy loss, which is standard in the SSL literature. One can use a more advanced supervised training e.g., compressing supervised ResNet50x4 to AlexNet, to get much better performance for the supervised model.

**Acknowledgment:** This material is based upon work partially supported by the United States Air Force under Contract No. FA8750-19-C-0098, funding from SAP SE, and also NSF grant number 1845216. Any opinions, findings, and conclusions or recommendations expressed in this material are those of the authors and do not necessarily reflect the views of the United States Air Force, DARPA, and other funding agencies. Moreover, we would like to thank Vipin Pillai and Erfan Noury for the valuable initial discussions. We also acknowledge the fruitful comments by all reviewers specifically by Reviewer 2 for suggesting to use teacher's queue for the student, which improved our results.

## Broader Impact

**Ethical concerns of AI:** Most AI algorithms can be exploited for non-ethical applications. Unfortunately, our method is not an exception. For instance, rich self-supervised features may enable harmful surveillance applications.

**AI for all:** Model compression reduces the computation needed in inference and self-supervised learning reduces annotation needed in training. Both these benefits may make rich deep models accessible to larger community that do not have access to expensive computation and labeling resources.

**Privacy and edge computation:** Model compression enables running deep models on the devices with limited computational and power resources e.g., IoT devices. This reduces the privacy issues since the data does not need to be uploaded to the cloud. Moreover, compressing self-supervised learning models can be even better in this sense since a small model e.g., MobileNet that generalizes to new tasks well, can be finetuned on the device itself, so even the finetuning data does not need to be uploaded to the cloud.

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
