[Supplementary Material]

# Appendix

Table A1: **Parameter and FLOPs comparison:** We report the number of floating point operations (FLOPs) and the number of parameters in a model below.

| Model | FLOPs (G) | Params (M) |
|---|---|---|
| MobileNet-V2 | 0.33 | 3.50 |
| AlexNet | 0.77 | 61.0 |
| ResNet-18 | 1.82 | 11.69 |
| ResNet-50 | 4.14 | 25.56 |
| ResNet-50x4 | 64.06 | 375.38 |

## More results for cluster alignment

For cluster alignment experiment (Section 4.3), we calculate the alignment for each category, sort them, and show in Figure A1. Moreover, Figure A2 is a larger version that is generated similar to Figure 3-right. Each row is a random cluster while images in the row are randomly sampled from that cluster with no manual selection or cherry-picking.

Figure A1: **Cluster alignment accuracy:** We calculate the accuracy for each ImageNet category and then plot them after sorting.

## Implementation details for the baselines

**Non-compressed (MoCo):** We use MoCo-v2 [10] from the official code [44] with AlexNet, ResNet-18, and MobileNet-V2 architectures for 200 epochs. We also train a longer baseline with ResNet-18 for 1000 epochs. All other hyperparameters are the same as the official version (m=0.999, lr=0.03).

**CRD:** We use the official code provided by the authors[12] and removed the supervised loss term. We use their default ImageNet hyperparameter of $lr = 0.05$ except for AlexNet student for which we use $lr = 0.005$ to make it converge.

**CC:** We calculate the $\ell_2$ normalized embeddings for the entire training dataset and apply k-means clustering with ($k = 16,000$) (which is adopted from [58]). This is equivalent to clustering with cosine similarity. We got slightly better results for cosine similarity compared to Euclidean distance. We use the FAISS GPU based k-means clustering implementation[1]. Finally, the student is trained to classify the cluster assignments. As in [38, 58], we train the student for 100 epochs. We use $lr = 0.1$ for ResNet models and $lr = 0.01$ for MobileNet-V2 and AlexNet models. We use cosine learning rate annealing.

**Reg:** We use Adam optimizer with weight decay of $1e-4$ for 100 epochs, and batch size of 256. For MobileNet-V2 and ResNet-18, we use $lr = 0.001$, and for AlexNet $lr = 0.0001$. The lr is reduced

by a factor of 10 at the 40-th and 80-th epochs. We use ADAM optimizer as performed better than SGD.

**Reg-BN:** It is similar to Reg except that we use SGD optimizer with $lr = 0.1$ instead of ADAM.

### Details of Places experiments (Section 4.4)

We perform adaptive max pooling to get features with dimensions around 9K, and train a linear layer on top of them. Training is done for 90 epochs with lr = 0.01, batch size = 256, weight decay = $1e - 4$, momentum = 0.9, and lr multiplied by 0.1 at 30, 60, and 80 epochs.

### Details of PASCAL experiments (Section 4.4)

For classification, we train a single linear layer on top of a frozen backbone. We use SGD with learning rate = 0.01, batch size = 16, weight decay = $1e - 6$, and momentum = 0.9. We train for $80,000$ iterations and multiply learning rate by 0.5 every $5,000$ iterations.

For object detection, we use SGD learning rate = 0.001, weight decay = $5e - 4$, momentum = 0.9 and batch size = 256. We train for $15,000$ iterations and multiply learning rate by 0.1 every $5,000$ iterations. The training parameters are adopted form [38] and the code from [20].

### Details of small data ImageNet experiments (Section 4.4)

We train a single linear layer on top a frozen backbone. We use SGD with learning rate = 0.05, batch size = 256, weight decay = $1e - 4$, and momentum = 0.9. We use cosine learning rate decay and train for 30 and 60 epochs for 10 percent and 1 percent subsets respectively. The subsets and training parameters are adopted from [9].

Figure A2: **Cluster Alignment:** Similar to Figure 3 (c), we show 20 randomly selected images (columns) from 30 randomly selected clusters (rows) for our best AlexNet modal. This is done with no manual inspection or cherry-picking. Note that most rows are aligned with semantic categories.