[Reviews · NeurIPS 2020]

Review 1

Summary and Contributions: This paper propose a model compression method for self-supervised learning by using a modified model distillation framework. They first compute the similarity between a query image and images in a memory bank. Then, the model minimizes the distance (KL divergence) of such similarity distribution between a big network such as ResNet50(x4) and a small network such as AlexNet. The experiment results show that such self-supervised model compression technique can improve the performance of small networks on different datasets and tasks.

Strengths: 1. A simple and effective self-supervised model compression framework. The model outperforms many other state-of-the-art methods. 2. Extensive experiments on different datasets and tasks.

Weaknesses: 1. In Tab. 2, I find that the CC (Cluster Classification) outperforms the model in the paper with the same ResNet50 teacher net. It would be better to compare the results under the same setting, from same teacher net. 2. The results of '4.4 Experiments Comparing Self-Supervised Methods' seems not demonstrate the effectiveness of the model compression method itself, since the model is trained from the best self-supervised learning teacher model. When the teacher self-supervised learning is better, the student model should be better as well. As shown in Tab. 3, we can see MoCo outperforms other baselines, so it is not the contribution of the model itself but the teacher network. 3. Since it is a model compression method, I think it would be better to compare the model parameter and FLOPS between the student and teacher networks.

Correctness: Yes

Clarity: A better layout for Tab. 3-6 would make the paper clearer. typos: Tab. 2, right column, ResNet-50, 'Simclr' should be 'SimCLR'.

Relation to Prior Work: Yes

Reproducibility: Yes

Additional Feedback:


Review 2

Summary and Contributions: The paper explored the idea of using knowledge distillation to compress the representations learned from a large model to a small one. This is a very natural idea given that larger models are doing well on self-supervised representation learning, and given the good representation, a small model can be trained to mimic the representation (or at least the behavior) of the large model. The paper has done a good job in exploring and analyzing the design choices of this space.

Strengths: + The paper does a detailed exploration and analysis of the design choices for the idea of knowledge distillation in self-supervised learning. It includes different losses, different network architectures, different evaluation methods (e.g. linear classifier on ImageNet, transfer to other tasks), and different hyper-parameters. While it maybe less novel, it shows a solid set of reference results for follow-up works. + One nice thing about the paper is that most claims are either backed up by results, or by quoting from other papers. It shows scientific rigor and to me this is an important sign of a well-written paper. + Self or unsupervised representation learning is an important topic of interest today, so definitely related to the community.

Weaknesses: - One important analysis that may be missing is the ablation of whether to have separate student/teacher queues or not. Since the teacher network is anyhow used to provide the target for the student network, I feel the negative keys can come from the same queue (either fixed and provided by the teacher network, or computed online by a key-encoder using the student network), this can also decouple the two changes made in the transition from *regression* baseline to the "ours" method, as regression does NOT need queues at all in my understanding, but "ours" changed both the loss form and the queue -- it would be good to know which part plays a more important role here, the loss or the queue. - I am not sure how to add BN to the teacher network (R182), since I assume the teacher network is already fixed and only provides features, isn't this similar to just whitening the features? - According to Table 2, CC seems highly competitive to ours (achieving on-part performance to "ours" when it is ResNet-50), please show the performance of Ours with R504x for a fair comparison.

Correctness: - I am not sure the momentum is needed in the key encoder, because different from contrastive learning frameworks like MoCo, here the task is actually more similar to normal *supervised* learning because the teacher network is already trained and provides a target. - Typo: Fig 3. caption -- AlexNet modal -> AlexNet model

Clarity: (Duplicate question, already answered in above)

Relation to Prior Work: * The paper did a good job in putting the current work in context of the literature, and I haven't seen similar works before, so it is good on this front.

Reproducibility: Yes

Additional Feedback:


Review 3

Summary and Contributions: This paper tries to tackle the problem of compressing a big unsupervised model to a small unsupervised model. The way to do unsupervised distillation is to transferring the probability distribution of the contrastive task from the teacher model to student model. While the authors keeps arguing that a self-supervised AlexNet surpasses supervised AlexNet, I find myself hard to buy this claim. I also think compressing unsupervised model now is meanless, as long as unsupervised methods are still evolving dramatically (see later). ===================== I thank the authors' feedback, and the rebuttal does address some of my questions. Therefore I would upgrade my score a bit. Meanwhile my biggest concern is still unsolved (Maybe I am wrong, but I am not convinced from the rebuttal): Whether the proposed method is compatible with different self-supervised learning methods? As we know these days SSL methods are evolving very fast, a method that are not necessarily compatible with different SSL methods is less attractive. If tomorrow a new SSL method yields a better R-50 (or whatever model) outperforming the distilled model here, we may directly use the model from the new method, rather than pretrain a big ResNet-50x4 with MoCo and then perform distillation proposed here, to just get an inferior model. Therefore, to me, it's important to demonstrate the proposed method is compatible with different SSL methods, so that it (potentially) could still help future SSL models. Since the distillation objective here is contrastive learning, whether it's compatible with different SSL methods is less clear to me. So my suggestion is to try to compress SSL models from different methods, which I think might help this paper attract wider attention.

Strengths: Firstly, the method of transferring the probability of contrastive task is simple and neat. The authors have conducted abundant empirical evaluation on a variety of tasks, including metric learning datasets CUB-200 and Car-196, ImageNet and Places datasets, Pascal VOC object detection.

Weaknesses: One big issue that I see is, it's not very meaningful to do model compression for unsupervised models before the current evolution of contrastive approaches plateau. - For example, you may distill from a big model to ResNet-50 to achieve a good performance today, then the next day a new contrastive paper comes out and achieves higher performance using only ResNet-50 without a teacher. Then why do we still need the distillation method proposed today, instead of directly using the new contrastive method? So this distillation method will quickly fade away. For instance, IIRC, [a] achieves 73.0% linear accuracy and transfer better to Pascal VOC and COCO, then the effectiveness of this paper will be largely discounted (by the way [a] might be discussed as well). - Distillation makes more sense in supervised setting because, you can always get performance boost for free, and that boost is non-trivial and can always be stacked to cross-entropy only models. But it's unclear how your method could improve upon better self-supervised methods, e.g., can you improve upon [a] using your method out of the box? [a] What makes for good views for contrastive learning I wonder if the proposed method could do self-distillation, i.e., you train a ResNet-50 by whatever self-supervised method, and then you trained a new ResNet-50 using your distillation method. For example, you can distill from ResNet-50 trained in [a] or MoCo v2 to see if you can improve upon it. - If this works, it would make your distillation method much more useful, as it could improve any new SoTA method that jump in in the future I did not buy the point of self-supervised AlexNet surpasses supervised one. - This is not surprising as you are distilling from a much bigger and better model. I can expect that once unsupervised ResNet-50 will surpasses supervised R-50 with distillation. - The self-supervised model is using heavier computation and longer training, making the comparison not head-to-head. Table 1 is a bit messy. Both supervised baseline, unsupervised baseline and different distilled models are put into the same section. Besides, whether the baseline methods have been carefully tuned is unclear. E.g., - Observation: while CC is much worse than CRD for AlexNet, it's much better for MobileNet-V2. Why? - Have you ever tuned different numbers of clusters for CC to make it optimal - Have you also tried to make every other components in CRD to be similar as your method here? E..g. non-linear projection head, as well as the number of negatives? For transferring to Pascal VOC, why not using ResNet-50, which is more interesting to see? Also, why not trying COCO? I might prefer COCO over PASCAL VOC.

Correctness: two wrong claim: Line 42-44, NN is not always faster than linear classifier in terms of inference. Imagine you have a very large training set, that computing pair-wise distances between a query point and those samples and do sorting is time-consuming as well. Line 79-81, these methods are also be applied to unsupervised compression, as long as you remove the standard cross-entropy term. For example, just transferring features.

Clarity: In general this paper is easy to follow.

Relation to Prior Work: I am generally OK with the related work, except for two very relevant papers are not discussed and included in Table 3: [1] Data-Efficient Image Recognition with Contrastive Predictive Coding [2] Contrastive Multiview Coding

Reproducibility: Yes

Additional Feedback: see above


Review 4

Summary and Contributions: The authors propose a method to distill a self supervised model into another model with small capacity. Their main contribution is a mechanism to transfer knowledge from a "heavy" teacher model to a "light" student model -- dubbed compression. The method uses a large memory (128,000 entries) to store teacher embeddings of the anchor images. During training, they generate teacher and student embeddings for a query image, compute its distance to all the anchor features in the memory bank and convert that to a probability distribution using softmax. Using KL divergence they train the student to match teacher's distribution. Teacher is alway fixed/frozen.

Strengths: * The authors compare their compression method to several existing compression methods (from SSLed teacher models) and directly compare to SSLed student models. * The proposed method is quite simple and works significantly better than other compression methods. * The authors describe implementation details such as caching teacher embeddings, using, memory bank framework based on Momentum Contrast paper which are critical to use this method in practice.

Weaknesses: * In Table 1 they show that they are able to improve over Alexnet Supervised baseline, but this doesn't hold for any of the other student models. I am not sure that this is because of their method. It likely due to the fact that AlexNet supervised baseline does not use all the tricks used to train newer architectures like Resnet / Mobilenet V2. * Novelty in the paper seems a bit limited.

Correctness: * In table 2, the proposed method is similar to CC[32] in performance when using the same teacher model. However, the. authors claim that they are better than all methods on line 198-199 by comparing to a model trained using Resnet50x4 teacher. IMO its a bit unfair.

Clarity: Yes, the paper is well written for most part. The tables in results section are a bit difficult to parse easily, but no other concerns.

Relation to Prior Work: Yes, the authors describe and differentiate their work other related works.

Reproducibility: Yes

Additional Feedback: Typo: line 122 -> shaper

[Author Response · NeurIPS 2020]

We thank the reviewers for their valuable comments.

**R1, R2, R3, R4: Summary and novelty:** Recent works have shown that the accuracy gap between supervised and
self-supervised (SSL) models can be dramatically reduced by increasing the model capacity. We propose a simple yet
effective unsupervised compression method for SSL representations that reduces the gap in less complex models. Our
experiments compare with other compression methods and also show that using our method along with a strong SSL
model outperforms all SOTA SSL methods. All reviewers agree that our method is simple, effective, and rigorously
evaluated against competent baselines on different datasets and tasks. Below, we address concerns from the reviewers.

**R1, R2, R4: Table 2:** We agree that highlighting in this
table is confusing. For a fair comparison, we did try CC
with R50x4 teacher. The results are shown in the table
beside. Our model outperforms CC when the teacher is
R50x4 and is on par with CC when the teacher is R50. We
will clarify this in the text.

| AlexNet | | CUB200 | | | | Cars196 | | | |
|---------|---------|------|------|------|------|------|------|------|------|
| Method | Teacher | R@1 | R@2 | R@4 | R@8 | R@1 | R@2 | R@4 | R@8 |
| CC | R50 | **23.2** | 32.5 | **45.1** | **58.2** | **23.7** | 31.4 | 41.1 | 52.4 |
| Ours | R50 | 23.1 | **33.0** | **45.1** | 58.0 | 23.6 | **32.8** | **42.9** | **54.9** |
| CC | R50x4 | 23.6 | 33.6 | 44.9 | 58.4 | 25.4 | 33.2 | 43.2 | 54.3 |
| Ours | R50x4 | **26.5** | **37.0** | **49.4** | **62.4** | **28.4** | **38.5** | **48.7** | **60.4** |

**R1: Results of Section 4.4:** Our goal is to show that our distillation method can compress SSL representations. In
Sec 4.3, we show that our method is better than previous compression methods. In Sec 4.4, we aim to quantify its
improvement over SOTA SSL models. We do not claim a stand-alone SSL method. Our method can improve upon other
SSL methods when they are used as teachers. **Tables 3-6:** We will improve the organization of those tables. **FLOPS:**
We will add FLOPS to the final paper. Compressing ResNet50x4 to AlexNet reduces computation by almost 80 times.

**R2: Ablation:** Yes, one can use the teacher's embedding for student's anchor points, but our goal was to decouple
the embedding spaces of the teacher and student so that they can easily use different architectures. But, it's a good
idea and we will add this experiment. Note that we show the effect of changing the queue size in Fig. 3.a. **BN layer:**
Yes, we simply whiten each dimension by the batch statistics using the implementation of BN module. We will clarify.
**Momentum:** Yes, we agree. Indeed, we show in Line 243 that momentum does not affect our results much.

**R3: "Compressing unsupervised model now is meanless, as long as unsupervised methods are still evolving
dramatically":** We respectfully disagree. All machine learning algorithms are evolving and we should not wait to
develop something novel. We believe there is some misunderstanding that our method is specifically designed for
contrastive learning (CL). This is not true. Our formulation looks similar to CL, but we never compare representations
of the same image. Instead of having positive or negative pairs, we compare each image with all "other" images, called
anchor points, and transfer the actual similarity values.

**R3: Surpassing supervised AlexNet:** As pointed out in Lines 252-255, we are not claiming that our SSL method
works better than supervised models "in general". We simply compare with the supervised AlexNet that is trained
with cross-entropy loss, which is standard in the SSL literature. One can use a more advanced supervised training e.g.,
compress supervised ResNet50x4 to AlexNet, to get much better performance for the supervised model. We will clarify
this more in the paper.

**R3: COCO and ResNet50:** Since our main focus was on compressing to smaller models, we used AlexNet for transfer
experiments, which is standard in self-supervised learning community. Note that transferring to COCO needs more
resources, and we have already done lots of experiments as acknowledged by other reviewers.

**R3: NN is not always faster than linear classifier:** In Lines 42-44, by "evaluation", we meant evaluating the SSL
features (training a linear model and testing it). NN is faster since it does not need any training and parameter tuning.
We will clarify in the final version. **Removing cross entropy loss from supervised distillation methods:** Yes, we
agree. This is exactly what we did when using CRD (SOTA method) in our experiments. Note that CRD [43] is cited
on Line 79. **Related work:** Thanks for pointing out those two papers. We missed them as they are not closely related
and are not peer-reviewed. We will add them to the final paper. **Citing [a]:** [a] is a concurrent work that appeared on
ArXiv just 17 days before our submission. We will cite it. **Self-distillation:** Our goal was to compress SSL models, so
we did not consider self-distillation. That can be an interesting future work.

**R3: Table 1 is a bit messy:** We will improve its organization. We chose number of clusters for CC from the analysis in
ClusterFit [49] and tuned the other parameters (learning rate, its schedule and number of iterations). CRD [43] already
has a non-linear projection head and the paper shows that it is not very sensitive to the number of negatives beyond
4,096. We do not know why MobileNet-V2 is different from AlexNet in comparing CC with CRD.

**R4: Why AlexNet is better:** Yes, we agree that our method does not improve over supervised version of other
architectures, but it consistently reduces the gap between supervised and unsupervised models. In general, the deeper
the teacher compared to the student, the more improvement from compression methods. In out experiments, this ratio is
maximized when we compress ResNet50x4 to AlexNet. We are using standard optimization for all architectures but
AlexNet does not benefit from recent architectural tricks like BN and Residual layers.

[Meta-Review · NeurIPS 2020]

This paper presents an approach for distillation of self-supervised models. All the reviewers acknowledge that the paper present a simple approach which outperforms several baselines. There are some concerns with respect to: (a) speed with which SSL field changes and applicability to new approaches; (b) clarity of tables; (c) claim of better than alexNet supervised. There was a rebuttal which answered some of the concerns. The AC agrees with authors that we should not wait for better models before working on model compression. Based on the discussions, the AC recommends acceptance.